# Real-time Hall-effect detection of current-induced magnetization dynamics in ferrimagnets

G. Sala [1✉], V. Krizakova [1], E. Grimaldi[1], C.-H. Lambert[1], T. Devolder[2] & P. Gambardella [1✉]

Measurements of the transverse Hall resistance are widely used to investigate electron transport, magnetization phenomena, and topological quantum states. Owing to the difficulty of probing transient changes of the transverse resistance, the vast majority of Hall effect experiments are carried out in stationary conditions using either dc or ac. Here we present an approach to perform time-resolved measurements of the transient Hall resistance during current-pulse injection with sub-nanosecond temporal resolution. We apply this technique to investigate in real-time the magnetization reversal caused by spin-orbit torques in ferrimagnetic GdFeCo dots. Single-shot Hall effect measurements show that the current-induced switching of GdFeCo is widely distributed in time and characterized by significant activation delays, which limit the total switching speed despite the high domain-wall velocity typical of ferrimagnets. Our method applies to a broad range of current-induced phenomena and can be combined with non-electrical excitations to perform pump-probe Hall effect measurements.

[1] Department of Materials, ETH Zurich, Zurich, Switzerland. [2] Centre de Nanosciences et de Nanotechnologies, CNRS, Université Paris-Sud, Université Paris-Saclay, Orsay Cedex, France. ✉email: giacomo.sala@mat.ethz.ch; pietro.gambardella@mat.ethz.ch

The broad family of Hall effects includes phenomena of ordinary, anomalous[1], planar[2,3], topological[4,5], and quantum[6-8] origin. These effects have become standard tools for benchmarking the physics of metallic, semiconducting, and topological materials as well as the functionality of electronic and spintronic devices. The anomalous Hall effect (AHE), for example, allows for probing the emergence of magnetically-ordered phases[1,9-11], field-[12] and current-induced magnetization reversal[13-15], domain-wall motion[16], and spin-orbit torques (SOTs)[17-19]. Measurements of the transverse resistance also provide insight into magnetoresistive phenomena, such as the planar Hall effect and spin Hall magnetoresistance, which can be used to track the response of antiferromagnets and magnetic insulators to applied magnetic fields, currents, and heat[20-22]. Extending these measurements to the time domain would enable access to the dynamics of a vast range of electronic and magnetic systems. As is well-known, the ordinary and planar Hall effects are widely employed in sensors for the detection of magnetic fields and microbeads[23-25], and have a frequency bandwidth extending up to several GHz[26,27]. However, there are only few examples of time-resolved (ns-μs) measurements of the magnetization dynamics using the Hall effect, which are limited to observations of laser-induced heating[28] and the transit of domain walls[29,30].

Here, we present an all-electrical technique suitable for systematic real-time measurements of any kind of transverse magnetoresistance in devices with current flowing in-plane. The key idea consists in disentangling the tiny magnetic Hall signal from the large non-magnetic background by minimizing the current leakage in the sensing arms of the Hall cross. This approach, which relies on the counter-propagation of electric pulses, is well adapted for radio-frequencies and proves particularly useful for fast excitations, e.g., ns- and sub-ns-long pulses. We demonstrate the capability of this technique by studying the magnetization dynamics triggered by SOTs[17] in ferrimagnetic GdFeCo dots patterned over a Pt Hall bar. In our detection scheme, the ns-long pulses do not only generate the perturbation on the magnetization, but also serve as the tool for tracking the magnetic response, including single-shot switching events. This capability opens up the possibility of performing systematic time-resolved Hall measurements of current-induced excitations in a broad variety of planar devices and provides access to stochastic events.

Ferrimagnets have recently attracted considerable attention due to the enhanced SOT efficiency[31-33] and the extraordinary high current-induced domain-wall velocity[34-36] attained, respectively, at the magnetization and angular-momentum compensation points. These properties make them promising candidates for the realization of fast and energy-efficient spintronic devices[36]. However, the current-driven magnetization dynamics in these systems has been investigated only using magneto-optical pump-probe methods[36,37], which do not provide information on stochastic events. Our time-resolved AHE measurements show that the reversal of the magnetization in GdFeCo evolves in different phases, which comprise an initial quiescent state, the fast reversal of the magnetization, and the subsequent settling in the new equilibrium state without ringing effects. Despite the high domain-wall velocity attained by ferrimagnets, we find that the total switching time is severely affected by an initial activation phase, during which the magnetization remains quiescent. We associate this phase, which has not been reported so far in ferrimagnets, with the time required to nucleate a reversed domain assisted by Joule heating. The single-shot AHE traces reveal the existence of broad distributions of the nucleation and reversal times and disclose the stochastic character of the SOT-induced dynamics, which is not accessible to pump-probe techniques. Our measurements further show that the domain nucleation time can be substantially reduced by increasing the current amplitude, leading to a minimum of the critical switching energy for pulses of reduced length.

## Results

**Time-resolved anomalous Hall-effect measurements**. Electrical time-resolved measurements using the Hall effect, or any form of transverse magnetoresistance, suffer from the difficulty of generating a detectable Hall signal without spoiling the signal-to-noise ratio. The main obstacle is the current shunting into the sensing line of the Hall cross, caused by the finite electric potential at its center. When a pulse reaches the cross, a portion of the current flows through the transverse arms (along $\pm y$ in the top panel of Fig. 1a), thus producing a spurious electric potential associated with the resistance of the leads. This potential is much larger than the signal of magnetic origin and hinders its detection. A limitation remains even in differential measurements because the unavoidable asymmetry of the leads introduces a finite differential offset[23] that can saturate the dynamic range of the Hall voltage amplification stage. These problems do not exist in standard dc measurements as the current leakage is countered by the high input impedance of the measuring instrument. At high frequency, however, impedance matching requires a low resistance (50 Ω) at the input port of the instrument, usually an oscilloscope.

The approach that we introduce here consists in injecting two counter-propagating rf pulses with amplitude $\left|\frac{V_P}{2}\right|$ and opposite polarity, as depicted in the bottom panel of Fig. 1a. Provided that these pulses reach the center of the cross at the same time and have the same amplitude, a virtual ground is forced there. The virtual ground limits the spread of the current because the voltage drop on the entire sensing line (Hall arm, cable, and input impedance of the oscilloscope) is ideally zero. The synchrony of the two balanced pulses, generated by a balun power divider, is ensured by the symmetry of the paths connecting the balun to the device, as schematized in Fig. 1b. Thanks to the opposite polarity of the pulses, the current flows along the $x$ direction, with double magnitude relative to the current produced by a single pulse of amplitude $\frac{V_P}{2}$, and sign determined by the polarity of the pulses. The current generates time-dependent transverse Hall voltages, $V_+$ and $V_-$, which are pre-amplified and acquired by a sampling oscilloscope triggered by an attenuated portion of the original pulse. If no change of the magnetization occurs during the pulses, the magnetic signal mimics the shape of the pulse. A deviation from this reference signal is the signature of ongoing magnetization dynamics. In the specific case discussed below, the transverse voltage stems from the AHE and its change over time gives access to the out-of-plane component of the magnetization. We note that, in the more general situation of asymmetric Hall crosses, our technique allows for compensating detrimental resistance offsets by tuning the relative amplitude of the counter-propagating pulses. This capability is unique to our approach and cannot be implemented in time-resolved differential Hall measurements[30]. We also remark that the main additional component to the setup required by our approach is the balun divider, which is a simple and affordable circuit element. More details about the electric circuit, including the rf and dc sub-networks, sensitivity, resistance offsets compensation, and time-resolution are discussed in the Methods and in Supplementary Notes 1, 2, and 5.

**Switching dynamics of ferrimagnetic dots**. We adapted this concept to investigate the SOT-induced magnetization switching of 15-nm-thick, 1-μm-wide $Gd_{30}Fe_{63}Co_7$ dots with perpendicular magnetization, patterned on top of a 5-nm-thick Pt Hall bar (see Fig. 1c, d, Methods, and Supplementary Note 3). The compensation temperature of the ferrimagnetic dots is below room

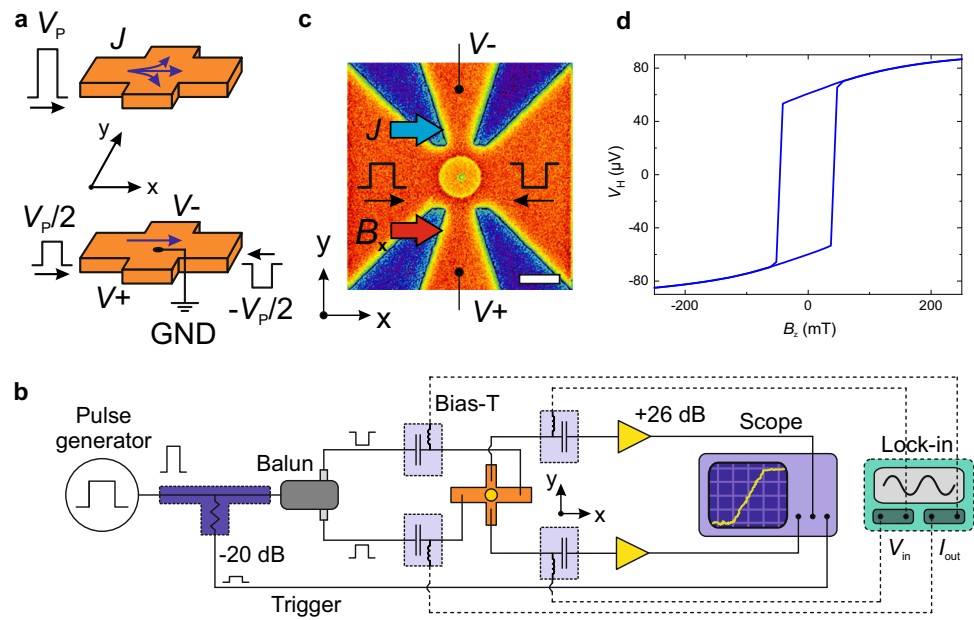

**Fig. 1 Experimental setup for time-resolved Hall-effect measurements. a** The injection in a Hall cross of a single pulse with amplitude $V_P$ causes current ($J$) shunting in the transverse sensing line (along $y$ in the upper panel). In contrast, two pulses with opposite polarity ($\frac{V_P}{2}$) that meet at the center of the Hall cross impose a virtual ground, thereby forcing the current to propagate along the main channel (along $x$ in the bottom panel). **b** Schematics of the rf setup. The initial pulse is fed to a balun divider, which splits the signal into two half pulses with opposite polarity that reach the device at the same instant. The current-induced transverse Hall potentials are amplified and detected by the oscilloscope, triggered by an attenuated portion of the initial pulse. Note that the electric paths traversed by $V_+$ and $V_-$ are symmetric and have equal length in the real setup. The dc sub-network (lock-in amplifier and bias-Ts, dashed lines) allows for the static characterization of the device. **c** The device is a 1-μm-wide ferrimagnetic GdFeCo dot at the center of a Pt Hall cross, as shown by the false-color scanning electron micrograph. The in-plane magnetic field $B_x$ is collinear to the current. The scale bar corresponds to 1 μm. **d** Out-of-plane hysteresis loop of a GdFeCo dot measured by the anomalous Hall effect.

temperature, such that the net magnetization and AHE are dominated by the magnetic moments of Fe and Co. Therefore, in our room-temperature measurements the current-induced switching in the presence of an in-plane static magnetic field has the same polarity as in perpendicularly magnetized ferromagnets with a Pt underlayer[14,17]. Specifically, the parallel alignment of current and field favors the down state of the magnetization, whereas the antiparallel orientation promotes the up state, which correspond to the negative and positive anomalous Hall resistance, respectively.

The differential signal $V_+ - V_-$ is determined by the magnetization orientation, which changes with time during a switching event. Figure 2a shows the switching traces obtained by measuring $V_+ - V_-$ during the reversal of a GdFeCo dot for different pulse amplitudes. In order to minimize spurious contributions to the magnetic signal, a background signal was recorded by fixing the magnetization in the initial state, either "up" or "down", and subtracted from the data. The down-up and up-down switching traces obtained by averaging over 1000 pulses are shown as red and blue lines, respectively. The black lines represent a reference trace obtained by subtracting two background measurements corresponding to the magnetization pointing up and down. This reference trace describes the maximum excursion of the Hall voltage during a current pulse (see Supplementary Note 4 for more details). The deviation of the switching traces from the top and bottom reference levels corresponds to the change of the out-of-plane magnetization driven by the SOTs during the 20-ns-long current pulse. Dividing the switching traces by the corresponding reference trace provides the normalized magnetic time traces shown in Fig. 2b–e. In these average measurements, the transition between the top and bottom reference levels of the switching trace is sufficiently clear such that the normalization by the reference trace is not strictly

required. The latter, however, is important to highlight the switching in single-shot measurements, which will be presented later on.

The measurements in Fig. 2b–e allow us to electrically probe the time-resolved SOT-induced dynamics in planar devices, which so far has been achieved only by X-ray and magneto-optical techniques[36–39]. We find that the switching dynamics of the ferrimagnetic dots comprises three phases: an initial quiescent state, the reversal phase, and the final equilibrium state, with the magnetization remaining constant both before and after the reversal. Both the quiescent and reversal phase present stochastic components. The observation of a long quiescent phase challenges the common assumption that the magnetization reacts instantaneously to the SOT owing to the orthogonality between the initial magnetization direction and the torque[40–42], unlike the spin-transfer torque between two collinear magnetic layers[43]. Instead, our measurements show that the duration of this phase can be comparable to the pulse length. The quiescent phase is a characteristic of the thermally activated regime, in which thermal fluctuations assist the switching and lead to a stochastic delay time. Because of the relatively high perpendicular anisotropy of the ferrimagnetic dots (see Supplementary Note 3), the thermal activation plays a role up to current density of the order of $1.5 \times 10^{12}$ A m$^{-2}$, similar to the switching of high-coercivity ferromagnetic nanopillars by spin-transfer torque[44]. By increasing the pulse amplitude or the in-plane field, the duration of the quiescent phase is significantly reduced as the switching dynamics approaches the intrinsic regime (see Fig. 2b–e and the following sections).

**Single-shot measurements.** Although the averaging process improves the quality of the traces, it conceals the stochastic nature

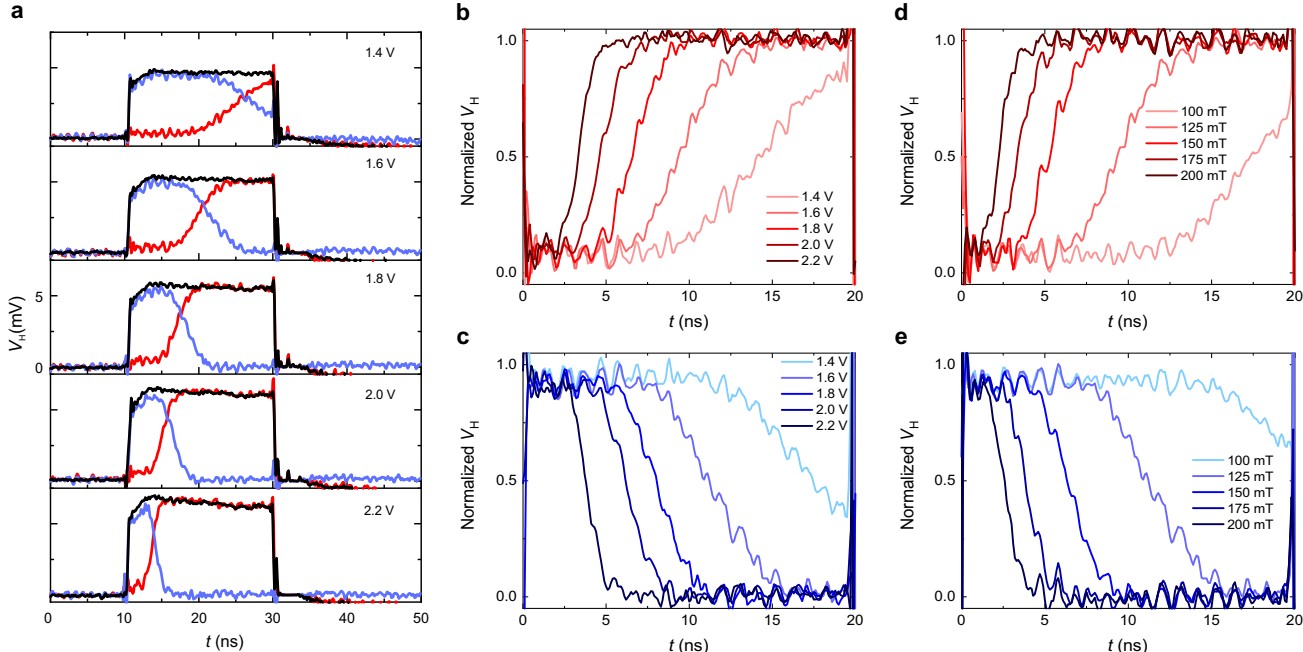

**Fig. 2 Switching dynamics of ferrimagnetic dots. a** Reference (in black) and switching traces of Pt/GdFeCo dots for 20-ns-long voltage pulses of increasing amplitude, showing up-down (blue lines) and down-up (red lines) reversals. The curves are averages of 1000 events. The in-plane magnetic field is 125 mT. **b**, **c** Normalized down-up and up-down switching traces at different pulse amplitudes corresponding to the traces in **a**. The current density in the Pt layer corresponding to a pulse amplitude of 1.4 V is $\approx 5.2 \times 10^{11}$ A m$^{-2}$. **d**, **e** Normalized down-up and up-down switching traces at different in-plane fields, for pulses with 1.6 V amplitude. In all the measurements the current was positive, whereas the field was positive (negative) in **c**, **e** (**b**, **d**).

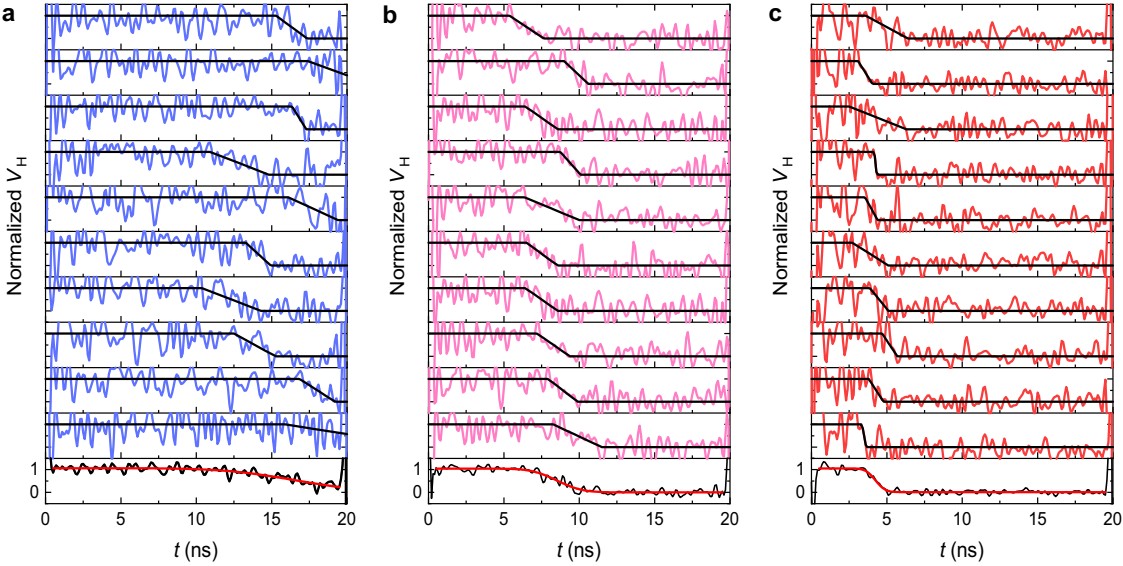

**Fig. 3 Single-shot Hall-effect measurements.** Normalized single-shot traces of Pt/GdFeCo dots for 20-ns-long pulses. The pulse amplitude is 1.4, 1.8, and 2.2 V in **a**, **b** and **c**, respectively. The in-plane magnetic field is 125 mT. The pulse amplitude in **a** is close to the threshold switching voltage (see Supplementary Note 3). The black lines are fits to the traces with a piecewise linear function. The bottom-most curve in each graph is the average of the 10 traces above, fitted with the cumulative Gaussian function (red).

of the dynamics. Here, we show that our technique provides sufficient signal-to-noise contrast to detect individual reversal events in Hall devices. By using the procedure outlined above, we measured single-shot switching traces for different in-plane magnetic fields and pulse amplitudes, as shown in Fig. 3 for three representative voltages. The single-shot traces are qualitatively similar to the average traces. However, the duration of the

quiescent and transition phases varies significantly from trace to trace. By fitting each trace to a piecewise linear function, we define $t_0$ as the duration of the initial quiescent phase during which the normalized Hall voltage remains close to 1 (0) before the up-down (down-up) reversal (see "Methods"). In the following, we refer to $t_0$ as the nucleation time, arguing that the quiescent phase is associated with the reversal of a seed

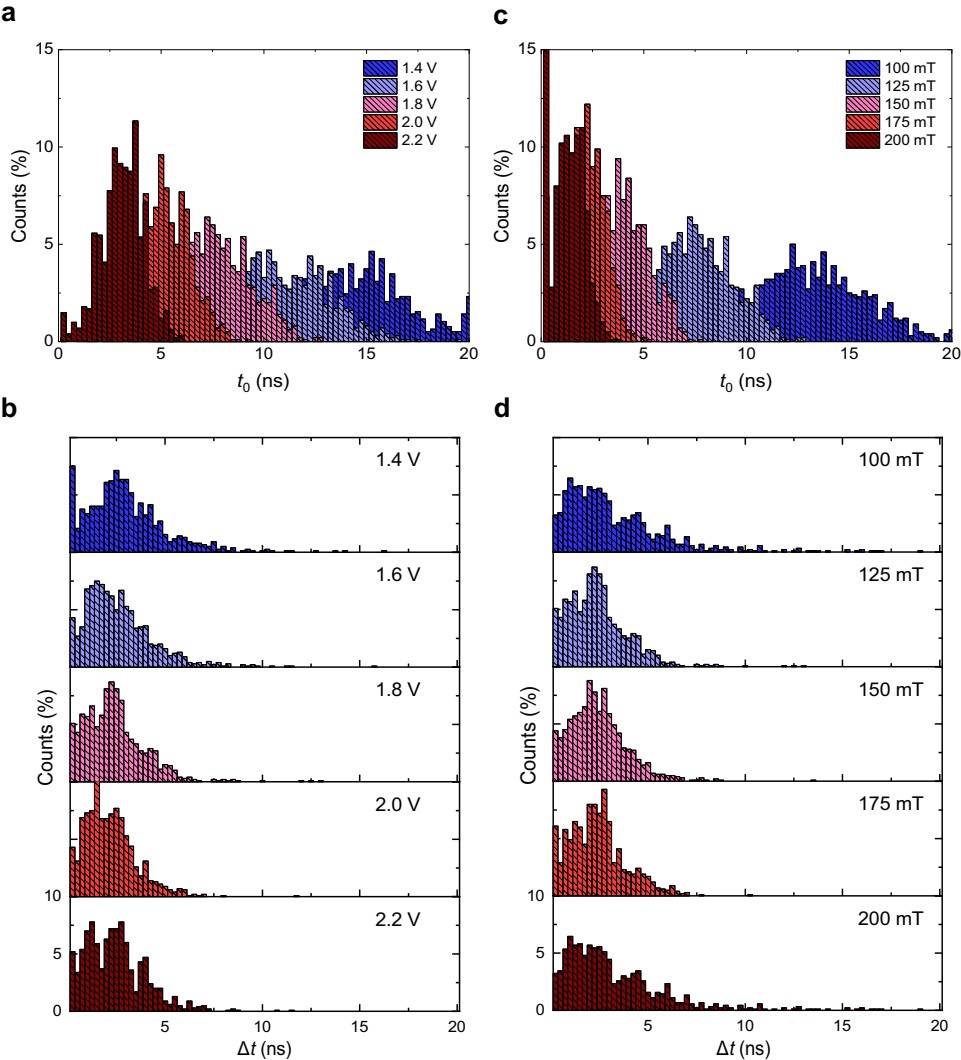

**Fig. 4 Distribution of nucleation and transition times. a, b** Percentage distributions of the nucleation time $t_0$ and transition time $\Delta t$ for different amplitudes of 20-ns long voltage pulses, extracted from the fits of the single-shot traces. At 1.4 V, the magnetization does not switch in 22.5% of the events; these events are not included in the plot. The in-plane field is 125 mT. **c, d** Same as **a, b** for different in-plane fields at a constant pulse amplitude of 1.8 V. At 100 mT, the magnetization does not switch in 9.8% of the events. At 200 mT, the left-most bin includes 24% of the events. This is likely an artifact of the fits due to the limited signal-to-noise ratio of the traces, which causes difficulties in fitting the dynamics close to the rising edge of the pulse. To ease the comparison, in all of the graphs the binning size is 250 ps, larger than the temporal resolution of 100 ps.

domain[38,45,46], in analogy to measurements performed on ferromagnetic tunnel junctions[47]. Additionally, we designate the duration of the transition between the up-down or down-up magnetization levels as the transition time $\Delta t$[48]. The total switching time is thus given by $t_0 + \Delta t$.

To gain insight into the stochastic variations of $t_0$ and $\Delta t$, we recorded a set of 1000 individual traces for several values of the applied in-plane field $B$ and voltage $V$. Figure 4 shows the statistical distributions of $t_0$ and $\Delta t$ obtained at representative fields and pulse amplitudes. The comparison between the single-shot statistics in Fig. 4 and the averaged traces in Fig. 2 reveals that the duration of the quiescent phase is systematically underestimated in the average measurements relative to the mean $\bar{t}_0$, whereas the duration of the transition phase is systematically overestimated relative to the mean $\overline{\Delta t}$. The deviation of the times deduced from the average measurements relative to $\bar{t}_0$ and $\overline{\Delta t}$ can reach up to −25% and 60%, respectively. The quantitative disagreement is determined by the superposition

of widely distributed nucleation events. As shown by the average curves at the bottom of Fig. 3, the large spread of the nucleation events anticipates the starting point of the average dynamics and, at the same time, broadens the apparent switching duration. Therefore, only single-shot measurements can accurately quantify the full switching dynamics, including the variability of events as well as the duration of the nucleation and transition phases, and their distributions.

The data reported in Fig. 4 show that $t_0$ approximately follows a normal distribution, as expected from random events. In contrast, $\Delta t$ has a significant positive skew with the mean $\overline{\Delta t}$ shifted towards the shorter times. Moreover, $\bar{t}_0$ and its standard deviation decrease strongly upon increasing either the pulse amplitude or the field, whereas $\overline{\Delta t}$ shows only a moderate dependence on the voltage. These distinct statistical distributions and dependencies are the signature of different physical processes underlying the initial phase and the transition phase of the reversal. Doubling the pulse amplitude or field leads to a ~10-fold

reduction of $\bar{t}_0$, consistently with an activated domain nucleation process that is promoted by SOTs and assisted by the in-plane field[47] and thermal fluctuations.

In contrast with $\bar{t}_0$, the effect of the in-plane field on $\overline{\Delta t}$ is negligible. This observation supports the interpretation of $\Delta t$ in terms of domain-wall depinning and propagation time, since, for the fields used in this study, the domain-wall mobility is saturated at the maximum value expected for Néel walls[49,50]. On the other hand, stronger pulses are expected to ease the depinning of domain walls and increase their speed, in accordance with the reduction of $\overline{\Delta t}$ at larger voltages. Consistent with our analysis, $\Delta t$ can be interpreted as the time required for the seed domain to expand across the entire area of the dot. Therefore, the inverse of $\Delta t$ provides an upper limit to the domain-wall velocity in our devices. The average domain-wall velocity estimated from the mean of the distributions reaches several hundreds of m/s, whereas the peak velocity can be as large as 4 km s$^{-1}$. Such a high speed is in line with the velocities estimated by measuring the domain-wall displacements in GdFeCo following the injection of current pulses[34–36]. Further improvements of the domain-wall velocities have been demonstrated by tuning the stoichiometry and transient temperature of GdFeCo so as to approach the angular-momentum compensation point[35]. Our measurements demonstrate that the nucleation phase, characterized by a long delay time $t_0$, is the real bottleneck of the SOT-induced switching dynamics of ferrimagnets. Therefore, the efficient operation of ferrimagnetic devices based on SOTs requires strategies to reduce the initial quiescent phase and mitigate the associated stochastic effects.

**Intrinsic and thermally activated switching regimes**. Measurements of the threshold switching voltage $V_c$ as a function of the pulse duration $t_P$ evidence the existence of two switching regimes[40], as shown in Fig. 5 (see also Supplementary Note 6). Above ~5 ns, $V_c$ changes weakly with $t_P$, which is a signature of the thermally-assisted reversal[40,44] and reveals the importance of thermal effects for the typical pulse lengths and amplitudes used in this study ($t_P = 2$ ns). On the other hand, the critical voltage increases abruptly for $t_P \lesssim 3$ ns, as expected in the intrinsic regime where the switching speed depends on the rate of angular-momentum transfer from the current to the magnetic layer. Indeed, in this regime, $V_c$ scales proportionally to $1/t_P$ (see Supplementary Fig. S7). Switching with $t_P = 300$ ps (equivalent

average domain-wall speed >3.3 km s$^{-1}$, under the assumption $t_0 \approx 0$) demonstrates that the quiescent phase can be suppressed by strongly driving the magnetization. In this case, the SOTs alone are sufficiently strong to drag the magnetization away from the equilibrium position and induce the nucleation of a domain against the energy barrier without substantial thermal aid. Finite element simulations support this point by showing that the temperature rise times in our devices are larger than 2 ns.

Importantly, the suppression of the quiescent phase requires more intense pulses but does not imply a larger energy consumption because the threshold energy density decreases by more than four times upon reducing $t_P$ from 20 ns to <1 ns (see Fig. 5). This favorable trend highlights the advantage of using materials for which the fast dynamics does not require excessively large current densities. We note that the current densities used in this study are compatible with previous results obtained on GdCo[36]. In that work, the current density at 300 ps is ~1.05 × 10$^{12}$ A m$^{-2}$, whereas in our devices with three times larger GdFeCo thickness the threshold current density reaches 3.6 × 10$^{12}$ A m$^{-2}$. For 20-ns-long pulses, this value reduces to 0.82 × 10$^{12}$ A m$^{-2}$. On the other hand, a more stringent comparison of our findings with the measurements reported in ref. [36] is not straightforward because the device geometries, the materials and their magnetic properties are dissimilar.

**Sensitivity and temporal resolution**. Finally, we present considerations on the sensitivity and time-resolution of our technique that apply to all conductors with a finite transverse resistivity $\rho_{xy}$. In all generality, we assume that $\rho_{xy} \neq 0$ only in a finite region of the Hall cross (the "magnetic dot"). The Hall voltage generated by two counter-propagating voltage pulses of opposite amplitude $V_P/2$ and $-V_P/2$ is given by $V_+ - V_- = f\frac{\rho_{xy}}{t}\frac{V_P}{R_I}$, where $t$ is the thickness of the dot, $R_I$ the resistance of the injection line, and $f$ a sensitivity factor (<1) that depends on the ratio between the area of the dot and the Hall cross as well as on the inhomogeneous current distribution within the device. An equivalent circuit model of the Hall cross and sensing apparatus shows that the differential Hall signal $S$ measured at the input ports of the oscilloscope is the result of the amplified voltage partition between the two branches of the sensing line, each having a resistance $R_S$, and the input resistance of the amplifier $R_A$:

$$S = 2G\frac{V_H}{2}\frac{R_A}{R_A + \frac{R_S}{2}}, \tag{1}$$

where $G$ is the gain of the amplifier stage. The total noise superimposed to the signal reads

$$N \approx 2\left(GN_{in} + 10^{\frac{NF}{10}}GN_{in} + \frac{10V_R}{2^8}\right), \tag{2}$$

where the first term represents the amplified sum of the Johnson and pulse generator noises ($N_{in}$), the second term the noise introduced by the amplifier with noise figure NF, and the third term the vertical resolution of the oscilloscope with 8 bits and acquisition range $V_R$ (see Supplementary Note 1 for a detailed derivation of Eqs. 1 and 2). On the basis of Eqs. 1 and 2, we estimate a signal-to-noise ratio $\frac{S}{N} \approx 2.2$ and $\approx 66$ for the single-shot and average traces measured with $V_P = 2.2$ V, respectively. These values are in fair agreement with the actual $\frac{S}{N}$ that characterizes the traces in Figs. 2 and 3. The main contributions to the noise are the NF of the amplifiers (54%) and the resolution of the oscilloscope (30%). The $\frac{S}{N}$ can thus be improved by means of amplifiers with lower NF (1–2 dB, against the 6 dB of our current setup) and oscilloscopes with higher vertical resolution (up to 10–12 bits) or better vertical range.

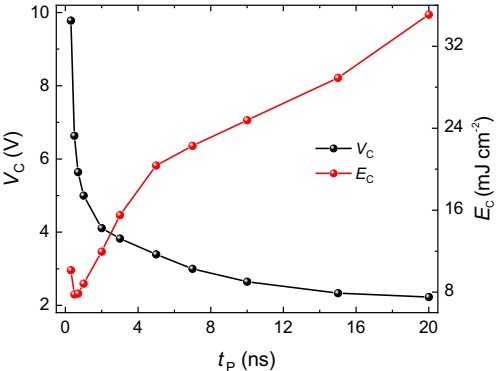

**Fig. 5 Switching with short pulses.** Threshold switching voltage (black dots, left scale) and energy density (red dots, right scale) as a function of the pulse length. The critical switching voltage is determined by after-pulse probability measurements as the voltage at which the device switches in 50% of the trials (see Supplementary Notes 3 and 6). The applied in-plane field is 100 mT.

The temporal resolution is determined by the sampling rate and bandwidth of the oscilloscope as well as by the acquisition mode. In this work, all the traces were acquired in the interpolated real-time mode, which allows for a nominal temporal resolution of $\approx 100$ ps, sufficient to track the dynamics of ns-long pulses. Using an oscilloscope with a higher sampling rate could improve the time-resolution down to about 10 ps. The minimal duration of the pulses that can be used to excite the magnetization, on the other hand, is determined by the impedance matching and symmetry of the circuit. In our case, the minimal pulse length is limited to a few ns by the inductive coupling between the wire bonds that connect the sample, which gives rise to over- and under-shoots in the transverse voltage at the rising and falling edges of a pulse (see Supplementary Note 4). This problem can be solved by using optimally-matched rf probes to connect the sample. Ultimately, it is of primary importance that the two branches of the injection (sensing) lines have equal lengths in order to guarantee the synchronization of the injected (sensed) signals. For symmetric branches, the relative delay of the balanced pulses at the center of the Hall cross is determined by the balun divider and is of the order of 1 ps[51]. Such a time lag limits the duration of the shortest measurable pulses.

## Discussion

We have demonstrated a technique to perform time-resolved measurements of the Hall effect and transverse magnetoresistive signals in devices with current flowing in-plane and applied it to investigate with sub-ns resolution the switching dynamics of ferrimagnetic dots induced by SOTs. Our results show that the current-induced magnetization reversal in GdFeCo is characterized by strong stochastic fluctuations of the time required to nucleate a domain. The quiescent phase that precedes the nucleation is a dynamical characteristic that ferrimagnets share with ferromagnets and that has not been reported previously for these materials. The observation of this phase, whose duration and variability are determined by the applied current and in-plane field, implies that the switching process is thermally activated. The corresponding switching delay depends on the combination of two effects. For a given strength of the SOTs and in-plane field, the average duration of the quiescent phase $\bar{t}_0$ is mainly determined by the temperature dependence of the magnetic anisotropy and the rate of increase of the temperature[47]. In this scenario, $t_0$ does not change between switching events and its standard deviation should be of the order of the pulse rise time. In addition to this deterministic process, $t_0$ is influenced by stochastic thermal fluctuations, which cause the spread reported in Fig. 4.

Upon reducing the length of the pulses and increasing their amplitude, the nucleation time can be suppressed to below 1 ns, which results in a minimum of the critical switching energy. Following the initial nucleation phase, the transition between two opposite magnetization states is both fast and monotonic, compatible with the extremely large domain-wall velocity reported for ferrimagnets. However, the reversal is also highly non-deterministic and characterized by a spread of transition times, which deserves further investigation. Overall, our data show that the switching delay time can be rather long in ferrimagnets, unlike the subsequent domain-wall motion, which is very fast. The coexistence of these slow and fast phases should be considered in future studies of ferrimagnets to correctly quantify the switching speed.

The sensitivity of the time-resolved Hall measurements is sufficient to perform both average and single-shot measurements, thus providing access to reproducible and stochastic processes. This dual capability combined with the straightforward implementation of our scheme and the widespread availability of Hall experimental probes makes our technique useful for a broad range of studies. The temporal evolution of the transverse voltage can be induced directly by the current, as in this work, or by a different stimulus, like magnetic fields, light or heat, using a pump-probe scheme with a variable delay time between excitation and counter-propagating voltage pulses. In the latter case, the electric current serves uniquely as the probing tool and its duration, amplitude, and waveform can be arbitrarily chosen. As any form of Hall effect or transverse magnetoresistance equally fits our detection scheme, potential applications include time-resolved investigations of electrically- and thermally-generated spin currents and spin torques in magnetic materials, switching of collinear and noncollinear antiferromagnets, as well as time-of-flight detection of skyrmion and domain walls in racetrack devices. Time-resolved Hall-effect measurements can also probe the emergence or quenching of symmetry-breaking phase transitions in driven systems. Further, as the Hall response is a quintessential signature of chiral topological states, real-time detection can provide insight into edge transport modes, as well as current-induced transitions between quantum Hall and dissipative states.

## Methods

**Device fabrication**. The Hall crosses and the dots were fabricated by lithographic and etching techniques. First, the full stack substrate/Ta(3)/Pt(5)/Gd$_{30}$Fe$_{63}$Co$_7$(15)/Ta(3)/Pt(1) (thicknesses in nm) was grown by dc magnetron sputtering on Si/SiN (200) substrate, pre-patterned by e-beam lithography, and subsequently lifted off. A Ti hard mask was defined by a second step of e-beam lithography, electron evaporation, and lift-off. The hard mask protected the circular areas corresponding to the dots during the Ar-ion milling that was used to etch the layers above Pt(5) and define the Hall crosses. Finally, Ti(5)/Au(50) contact pads were fabricated by optical lithography and electron evaporation, followed again by lift-off.

**Electrical setup**. With reference to Fig. 1, the pulses are produced by a reverse-terminated pulse generator (Kentech RTV40) with variable pulse length (0.3–20 ns, rise time <0.3 ns) and adjustable polarity, and fed to a directional coupler, which delivers a small portion (−20 dB) of the signal directly to the oscilloscope (trigger). The balanced-unbalanced (balun) power divider (200 kHz–6 GHz, Marki Microwave BAL-0006) splits the signal into two balanced pulses, with very similar amplitude. Next, the pulses travel to the Hall cross through identical paths. The four bias-Tees next to it combine the rf and dc sub-networks of the circuit, allowing both time-resolved (oscilloscope) and static (lock-in amplifier) measurements. Prior to detection, the transverse Hall potentials are amplified by amplifiers (Tektronik PSPL5865) with 26.5 dB voltage gain, 30 ps rise time and 30 kHz–12 GHz bandwidth. The oscilloscope is also a Tektronik instrument, with 2.5 GHz bandwidth, 20 GSa s$^{-1}$ sampling rate, and 50 Ω ac-coupled input impedance. A lock-in amplifier (Zurich Instruments MFLI) generates a small low-frequency sinusoidal current ($I_{out}$, 100–200 µA, 10 Hz) and demodulates the corresponding static anomalous Hall voltage ($V_{in}$). The Hall cross lies on a custom-built printed-circuit board with SMA connections and is contacted electrically by Al wire bonds. The device is located between the pole pieces of an electromagnet, whose magnetic field $B$ can be varied in amplitude and direction within the $xz$ plane.

**Fits of the time-resolved Hall voltage traces**. We fit the individual normalized switching traces with a piecewise linear function of the form:

$$\text{UP} - \text{DOWN} : y(t) = \begin{cases} 1, & t < t_0 \\ 1 - \frac{t - t_0}{\Delta t}, & t_0 < t < t_0 + \Delta t \\ 0, & t > t_0 + \Delta t \end{cases}$$

$$\text{DOWN} - \text{UP} : y(t) = \begin{cases} 0, & t < t_0 \\ \frac{t - t_0}{\Delta t}, & t_0 < t < t_0 + \Delta t \\ 1, & t > t_0 + \Delta t \end{cases}$$

for up-down and down-up switching, respectively. We chose a piecewise linear function because of its simplicity and its robustness with respect to the fitting routine as opposed to, e.g., the cumulative function of the Gaussian distribution, which is more prone to errors for small values of $t_0$.

## Data availability

The datasets generated and/or analyzed during the current study are available from the corresponding authors on reasonable request. The data for all of the figures are also

available in https://www.research-collection.ethz.ch/, https://doi.org/10.3929/ethz-b-000460625.

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

## Acknowledgements

This work was funded by the Swiss National Science Foundation (Grants Nos. 200020-172775 and PZ00P2-179944), the Swiss Government Excellence Scholarship (ESKAS-Nr. 2018.0056) and the ETH Zurich (Career Seed Grant SEED-14 16-2).

## Author contributions

P.G., E.G., G.S., and T.D. conceived the experiments. G.S and V.K. developed the setup and the measurement protocol. C.-H.L. deposited the samples. G.S. fabricated the device, performed the measurements, and analyzed the results. G.S. and P.G. wrote the manuscript. All authors discussed the data and commented on the manuscript.

## Competing interests

The authors declare no competing interests.
