## [Peer Review File · Nature Communications]

Reviewers' Comments:

Reviewer #2:

Remarks to the Author:

Sala et al. have responded to each of the points that I made in their previous version of manuscript. I see that the manuscript is now much improved. And now I can accept the novelty of their measurement technique. However, several concerns still remain as follows. To meet the criteria of *Nature Communications*, the following points should be clarified.

1. Origin of quiescent phase

In magnetization switching, the dynamic regime is generally classified into either thermally activated regime where the switching is governed by the thermal activation process over energy barrier when the driving current is lower than the threshold value or precessional regime where the switching is dominated by the angular momentum transfer when the driving current is higher than the threshold value. It is natural that the dynamic regime gradually changes from thermally activated regime to precessional regime as one increases the current magnitude. Indeed, the authors observed this dynamic regime transition in Fig. 5 of revised manuscript or Fig. R2(c) of rebuttal letter. From their results, I can see that the precessional regime can be obtained for $V > 4V_c$ where V_c and $1/t_p$ is linearly proportional to each other.

Meanwhile, as far as I know, the quiescent phase (or incubation time) is defined independent of dynamic regime. In spin transfer torque (STT)-driven magnetization switching, the incubation time generally originates from the precessional motion of magnetization with small thermal fluctuation. That is, the incubation time is generally defined in precessional regime, and is less dependent on the thermal activation process over the energy barrier.

In this manuscript, the authors mentioned that the incubation time in their ferrimagnetic sample originates from the thermal fluctuation. This means that what they measured is not the "real incubation time" but the "thermal activation time" over the energy barrier. This can be evidenced from the fact that the driving voltages that they used are below 2.2V, which corresponds in the thermal activation regime.

For me, it is quite nature to see the long and stochastic quiescent phase, because it is caused by the thermal activation process.

In this respect, I think the authors' claim that "The observation of a long quiescent phase challenges the common assumption that SOT-induced switching, unlike spin-transfer torque switching, has a negligible incubation time..." is quite misleading. In my opinion, the long quiescent phase is not from the inherent characteristics of SOT or ferrimagnet, but from the merely thermal activation process. The authors need to clarify this point.

2. Ferrimagnetic dynamics

Their answer to my question is unsatisfactory. They mentioned in their rebuttal that "the existence of the quiescent phase in ferrimagnets is somewhat unexpected", but I think it is 'expected', because the quiescent phase is not from the characteristics of ferrimagnet, but from the general

thermal activation process, as I discussed above. I think the same quiescent phase can be observed even they use ferromagnet, as long as the measurement regime is thermally activated regime. Are there any new findings related the antiparallel alignment of two sub-magnetic moments? For example, the angular momentum transfer between two sub-lattices as discussed in [Cai et al., Nat. Electron. 3, 37 (2020)]. They should clearly discuss why they used ferrimagnet and what they found as a something new originating from ferrimagnetic characteristics.

Reviewer #3:

Remarks to the Author:

This paper presents a clever Hall effect method for time-resolved measurements of current-induced switching in sub-micron-scale patterned magnetic dots. The effective removal of a spurious background, with the use of simultaneously injected opposite current pulses, permits this study to take advantage of the full dynamic range of an oscilloscope. This then allows for electrically detecting magnetic switching on the single-digit ns time scale with an unprecedented signal-to-noise ratio.

It is clear that study has been conducted with care, and the paper is very well written. I can easily recognize two high-impact, noteworthy contributions of this work.

1. Demonstration of the ability to acquire time-resolved, *single-shot* magnetic switching traces via the unique Hall measurement method. I believe this is a landmark advance that certainly benefits the fast growing field of spintronics (spinorbitronics). I also agree with the authors' claim that their time-resolved Hall method may find applications in other areas of condensed matter physics. I think that the authors make a good case that this method can be readily adapted by other laboratories.

2. Clear demonstration of the switching statistics of a ferrimagnet driven by current pulses. There has been a lot of interest in ferrimagnets lately, motivated by exciting reports of fast antiferromagnetic-like dynamics in these materials. To the best of my knowledge, this study is the first ever to present clear experimental insights into the timescales of current-induced switching in ferrimagnets, certainly by electrical means (as opposed to optical means). While some readers might find the presence of an incubation period to be not surprising, I think there is still significant merit in directly showing that the incubation period is rather long in these ferrimagnets, which are often associated with "fast" dynamics in researchers' minds. (I personally found it interesting/somewhat surprising to learn that there is several ns of incubation delay even when the subsequent domain wall motion, driven by a moderate current, is apparently quite fast.) The authors further show that the incubation delay is suppressed with a large enough current pulse. The result in Figure 5 is also insightful and encouraging for practical applications in that a larger threshold current pulse with a shorter duration actually leads to a smaller threshold energy to switch.

I believe that the above points are more than sufficient to justify the novelty and quality required for publication in Nature Communications. Particularly with the nice revision that the authors have already made, I do not have any significant points of criticism.

I only have two very minor comments.

* [This is an optional point, which I don't think necessarily needs to be addressed deeply.] How close is this GdFeCo film to (room temperature) magnetic or angular momentum compensation composition? Unless I missed something, the only information provided seems to be "The compensation temperature of the ferrimagnetic dots is below room temperature, such that the net magnetization and AHE are dominated by the magnetic moments of Fe and Co." It may be interesting to see a future work that studies how the composition affects the nature of switching, using the time-resolved method demonstrated here.

* Page 3, Line 9: Perhaps "setting" should be "settling"?

Reviewer #2

We thank the Referee for acknowledging the improvement of our manuscript and giving us the possibility of a further clarification. Please, find below our reply.

1. Origin of quiescent phase.

In magnetization switching, the dynamic regime is generally classified into either thermally activated regime where the switching is governed by the thermal activation process over energy barrier when the driving current is lower than the threshold value or precessional regime where the switching is dominated by the angular momentum transfer when the driving current is higher than the threshold value. It is natural that the dynamic regime gradually changes from thermally activated regime to precessional regime as one increases the current magnitude. Indeed, the authors observed this dynamic regime transition in Fig. 5 of revised manuscript or Fig. R2(c) of rebuttal letter. From their results, I can see that the precessional regime can be obtained for $V > 4V$ where V_c and $1/t_p$ is linearly proportional to each other. Meanwhile, as far as I know, the quiescent phase (or incubation time) is defined independent of dynamic regime. In spin transfer torque (STT)-driven magnetization switching, the incubation time generally originates from the precessional motion of magnetization with small thermal fluctuation. That is, the incubation time is generally defined in precessional regime, and is less dependent on the thermal activation process over the energy barrier.

In this manuscript, the authors mentioned that the incubation time in their ferrimagnetic sample originates from the thermal fluctuation. This means that what they measured is not the “real incubation time” but the “thermal activation time” over the energy barrier. This can be evidenced from the fact that the driving voltages that they used are below 2.2V, which corresponds in the thermal activation regime.

For me, it is quite nature to see the long and stochastic quiescent phase, because it is caused by the thermal activation process.

In this respect, I think the authors’ claim that “The observation of a long quiescent phase challenges the common assumption that SOT-induced switching, unlike spin-transfer torque switching, has a negligible incubation time...” is quite misleading. In my opinion, the long quiescent phase is not from the inherent characteristics of SOT or ferrimagnet, but from the merely thermal activation process. The authors need to clarify this point.

We agree with the Referee that the dynamics caused by low-amplitude pulses belongs to the thermally-activated regime, where the switching is triggered by the thermal activation over an energy barrier. The duration of the phase that precedes the switching (“quiescent phase”) depends on the combination of two effects, both of thermal origin. For a given strength of the spin-orbit torques, assisting magnetic field, and magnetic anisotropy, the average duration of the quiescent phase is determined by the dependence on temperature of the magnetic parameters and the rate of increase of the temperature. The magnitude of these parameters (saturation magnetization, magnetic anisotropy, magnetic stiffness, Dzyaloshinskii–Moriya interaction) keeps on decreasing with time until the switching threshold is reached, and the reversal begins by the formation of a seed domain. In this scenario, the duration of the quiescent phase does not change between switching events and, thus, the spread of the time t_0 should be negligibly small. In addition to this deterministic process, however, t_0 is influenced by thermal fluctuations, which introduce a source of stochasticity into the dynamics and eventually cause the spread of t_0 . In this sense, we have considered the latter as an incubation time, in analogy to the dynamics of the spin-transfer torques, where the incubation time is not necessarily associated with the precession of the magnetization but also to the formation of a reversed region (see e.g. Devolder, Physical Review Letters 100, 057206, 2008). Since we realize now that this terminology can cause confusion, we have replaced the expression “incubation time” with “thermal activation time” or “nucleation time” in the manuscript.

Therefore, we agree with the Referee that “the long quiescent phase does not come from the inherent characteristics of SOT or ferrimagnet, but from the merely thermal activation process”, provided that

this process includes the two components described above. It is important to notice, though, that for a long time it has been thought that the dynamics of spin-orbit torques does not comprise any quiescent phase, an assumption that was supported by measurements of several groups (see Refs. 38,39 and 41-43 as well as Yoon, *Science Advances* 2017;3: e1603099, and Sato, *Nature Electronics* 1, 508-511, 2018). Only recently, our time-resolved measurements in magnetic tunnel junctions and micromagnetic simulations have disclosed the role of the deterministic thermal activation that underlies the spin-orbit torque dynamics in ferromagnets when the current density is comparable to the threshold value (see Ref. 47). The data presented in this work show that ferrimagnets are severely affected by switching delays even at a current density well above the switching threshold (there is no precession in our case). In order to clarify this issue, we have modified the discussion of the quiescent phase in the manuscript.

For example, as far as we know, there is no report that, on the basis of simulations, predicts the existence of a quiescent phase. In all of the simulations, whether macrospin or micromagnetic, either the magnetization starts the switching instantaneously above the threshold current or it remains still. This is, indeed, because the importance of thermal effects close to the switching threshold conditions has never been considered. The data presented in this work confirm our recent findings (Ref. 47), but in a different device and, more importantly, in a different magnetic material.

We have therefore added a few sentences in the manuscript to stress the thermal origin of the quiescent phase in the thermally-activated regime.

2. Ferrimagnetic dynamics. Their answer to my question is unsatisfactory. They mentioned in their rebuttal that “the existence of the quiescent phase in ferrimagnets is somewhat unexpected”, but I think it is ‘expected’, because the quiescent phase is not from the characteristics of ferrimagnet, but from the general thermal activation process, as I discussed above. I think the same quiescent phase can be observed even they use ferromagnet, as long as the measurement regime is thermally activated regime. Are there any new findings related the antiparallel alignment of two sub-magnetic moments? For example, the angular momentum transfer between two sub-lattices as discussed in [Cai et al., *Nat. Electron.* 3, 37 (2020)]. They should clearly discuss why they used ferrimagnet and what they found as a something new originating from ferrimagnetic characteristics.

We measured the dynamics of ferrimagnets because these materials have raised a great interest in recent years, but the measurements of their actual current-induced dynamics is limited to the work of Cai et al. This study reports statistically averaged time-resolved data of domain wall transit times, which provide information on the reproducible dynamics of domain walls, but do not provide insight into the initial switching phase nor into stochastic phenomena. In contrast to ferrimagnets, the dynamics of ferromagnets has been deeply investigated by us and several other groups by means of electrical, optical, and x-ray techniques. For us, this is a sufficient justification for concentrating the attention on ferrimagnets, whose current-induced dynamics is still not fully known. In general terms, one may say that thermal activation is a phenomenon that plays a role in all processes of magnetization reversal at finite temperature. However, this has to be demonstrated and characterized. Similar to earlier measurements in ferromagnets, the recent reports on the ferrimagnetic dynamics do not provide insight into the regime where thermal effects play an important role. The binomial ferrimagnets-high speed is well established now, but this very fast dynamics only concerns the domain wall motion. As also recognised by Referee 3, it is interesting that the overall switching dynamics is the combination of a quite slow activation process and the rapid magnetization reversal. Identifying the existence of both phases is important to understand future measurements, for example to avoid underestimating the actual transition speed when the duration of the overall dynamics is mostly determined by the long quiescent phase.

Our measurements cannot provide a direct insight into the relative dynamics of the two Gd and FeCo sub-lattices and the possible transfer of angular momentum since the magneto-transport properties of

ferrimagnets are dominated by the transition metal. The same limitation concerns optical measurements. In this sense, we note that the discussion made by Cai et al. is based only on simulations. A deeper understanding of the role of the two sub-lattices and the influence of the compensation points may be obtained by performing a composition-dependent study. This is out of the scope of the current work but is certainly of interest for future studies.

Last, we would like to stress the novelty of our time-resolved technique. Ferrimagnetic dots were chosen to demonstrate the technique on a system with fast dynamics while providing at the same time much needed insight into their actual switching behavior.

Reviewer #3

This paper presents a clever Hall effect method for time-resolved measurements of current-induced switching in sub-micron-scale patterned magnetic dots. The effective removal of a spurious background, with the use of simultaneously injected opposite current pulses, permits this study to take advantage of the full dynamic range of an oscilloscope. This then allows for electrically detecting magnetic switching on the single-digit ns time scale with an unprecedented signal-to-noise ratio.

It is clear that study has been conducted with care, and the paper is very well written. I can easily recognize two high-impact, noteworthy contributions of this work.

1. Demonstration of the ability to acquire time-resolved, *single-shot* magnetic switching traces via the unique Hall measurement method. I believe this is a landmark advance that certainly benefits the fast growing field of spintronics (spinorbitronics). I also agree with the authors' claim that their time-resolved Hall method may find applications in other areas of condensed matter physics. I think that the authors make a good case that this method can be readily adapted by other laboratories.

2. Clear demonstration of the switching statistics of a ferrimagnet driven by current pulses. There has been a lot of interest in ferrimagnets lately, motivated by exciting reports of fast antiferromagnetic-like dynamics in these materials. To the best of my knowledge, this study is the first ever to present clear experimental insights into the timescales of current-induced switching in ferrimagnets, certainly by electrical means (as opposed to optical means). While some readers might find the presence of an incubation period to be not surprising, I think there is still significant merit in directly showing that the incubation period is rather long in these ferrimagnets, which are often associated with "fast" dynamics in researchers' minds. (I personally found it interesting/somewhat surprising to learn that there is several ns of incubation delay even when the subsequent domain wall motion, driven by a moderate current, is apparently quite fast.) The authors further show that the incubation delay is suppressed with a large enough current pulse. The result in Figure 5 is also insightful and encouraging for practical applications in that a larger threshold current pulse with a shorter duration actually leads to a smaller threshold energy to switch.

I believe that the above points are more than sufficient to justify the novelty and quality required for publication in Nature Communications. Particularly with the nice revision that the authors have already made, I do not have any significant points of criticism.

I only have two very minor comments.

*** [This is an optional point, which I don't think necessarily needs to be addressed deeply.] How close is this GdFeCo film to (room temperature) magnetic or angular momentum compensation composition? Unless I missed something, the only information provided seems to be "The compensation temperature of the ferrimagnetic dots is below room temperature, such that the net**

magnetization and AHE are dominated by the magnetic moments of Fe and Co.” It may be interesting to see a future work that studies how the composition affects the nature of switching, using the time-resolved method demonstrated here.

*** Page 3, Line 9: Perhaps “setting” should be “settling”?**

We thank the Reviewer for his/her appreciation of our work and the recognition of its quality and potential impact.

The GdFeCo devices studied here belong to a batch of samples with variable Gd concentration that cross the magnetization compensation temperature. However, we found that the fabrication steps alter the properties of the devices with respect to those of the full films. This undesired change is one of the limitations of amorphous ferrimagnets, which are particularly sensitive to standard operations such as the ion milling and the resist baking. These issues have already been observed by other groups (see e.g., Le Guyader, APL 101, 022410, 2012, or El-Ghazaly, APL 4, 232407, 2019, or Kirk, PR Materials 4, 074403, 2020) and are possibly caused by the selective oxidation or migration of the rare-earth atoms (see Hansen, Magnetic amorphous alloys, *Handb. Magn. Mater.* 6, 1991). Our estimate, based on the variation of the magnetization compensation temperature with the Gd concentration (about 30 K every 1%), is that the magnetization compensation temperature is around 250 K. Because of Joule heating during pulsing, we are also confident that our devices are always “FeCo-like”. We have added this information in the Supplementary Information 3.

Performing time-resolved measurements in samples that cross the compensation temperature is our current aim. We are therefore working on the fabrication procedure to avoid the mentioned side effects.

Reviewers' Comments:

Reviewer #2:

Remarks to the Author:

They have responded to all the points that I have made in the previous round, and revised the manuscript accordingly. I therefore recommend the publication of this manuscript.

Reviewer #3:

Remarks to the Author:

The authors have certainly addressed my comments in a satisfactory manner. In my view, the authors have addressed the concerns brought up by Reviewer #2 satisfactorily as well.

I believe this paper is suitable for publication in Nature Communications.